# An information-theoretic framework for deciphering pleiotropic and noisy biochemical signaling

Tomasz Jetka[1], Karol Nienałtowski[1], Sarah Filippi[2], Michael P.H. Stumpf [3] & Michał Komorowski[1]

Many components of signaling pathways are functionally pleiotropic, and signaling responses are marked with substantial cell-to-cell heterogeneity. Therefore, biochemical descriptions of signaling require quantitative support to explain how complex stimuli (inputs) are encoded in distinct activities of pathways effectors (outputs). A unique perspective of information theory cannot be fully utilized due to lack of modeling tools that account for the complexity of biochemical signaling, specifically for multiple inputs and outputs. Here, we develop a modeling framework of information theory that allows for efficient analysis of models with multiple inputs and outputs; accounts for temporal dynamics of signaling; enables analysis of how signals flow through shared network components; and is not restricted by limited variability of responses. The framework allows us to explain how identity and quantity of type I and type III interferon variants could be recognized by cells despite activating the same signaling effectors.

[1] Institute of Fundamental Technological Research, Polish Academy of Sciences, Warszawa 02-106, Poland. [2] Department of Mathematics and School of Public Health, Imperial College London, London SW7 2AZ, UK. [3] Melbourne Integrative Genomics, School of BioSciences and School of Mathematics and Statistics, University of Melbourne, Parkville 3010 VIC, Australia. Correspondence and requests for materials should be addressed to M.K. (email: m.komorowski@sysbiosig.org)

Biochemical signaling is a key mechanism to coordinate an organism in all aspects of its function. In a typical example, cells detect extracellular stimuli (input), e.g., growth factors, cytokines, or chemokines, with specific transmembrane receptors binding a ligand, which results in a biochemical activity on the inside of the cell, for example, the activation of a receptor-associated kinase[1]. The initial stimuli are processed in an intracellular relay mechanism and culminate in effectors (output), which might be transcription factors. The effectors carry information about the identity and intensity of the stimuli in order to initiate distinct cellular responses, which might involve gene transcription, or any other cellular process. Biochemical descriptions do not directly lead to understanding how the stimuli are translated into distinct responses as signaling processes are immensely complex[2,3]. Many components of signaling pathways are functionally pleiotropic:[4–6] (i) a single stimulus often activates multiple effectors, (ii) a distinct effector can be activated by numerous stimuli, and (iii) signals triggered by different stimuli often travel through shared network components. Besides, (iv) biochemical signaling processes are intrinsically stochastic and responding cells exhibit quite varied behaviors when examined individually[7,8], and (v) temporal dynamics of signaling in individual cells is correlated with physiological responses[6,9,10]. In light of these observations, understanding of how information about a complex mixture of extracellular stimuli is processed and translated into distinct cellular responses remains deficient[2,3]. For instance, human type I and type III interferons (IFNs) signal through distinct cell-surface receptors that appear to induce a shared signaling pathway. Yet, they can evoke different physiological effects[11]. The mechanism mediating this differential activity and signaling through common pathways remains largely unknown[11–15].

Understanding how cellular signaling processes can derive a variety of distinct outputs from complex inputs appears to be beyond solely experimental treatment. Therefore, an adequate modeling formalism is required. Following Berg and Purcell[16], probabilistic modeling has been applied to examine fidelity of receptors as well as more complex biochemical signaling systems[19]. Specifically, information theory has been deployed as an integrated measure of signaling accuracy, a term known as information capacity, $C^*$. Information capacity is expressed in bits, and generaly speaking, $2^{C^*}$ represents the maximal number of different inputs that a system can effectively resolve (e.g. different ligand concentrations)[17]. So far, both experimental and computational analysis of biochemical signaling within information theory revealed several unique aspects of how signaling pathways transmit information[18–24]. A tangible obstacle to further utilize the potential of information theory is the lack of computationally efficient tools that can account for complexities of biochemical signaling. Existing techniques are based on Blahut–Arimoto algorithm[18,25], small noise approximation[19,26], or density estimation[22] and their application so far has been limited to relatively simple systems, usually with one input and one output only. As analysis of systems with multiple inputs and outputs appears to be essential for deciphering of biochemical signaling[27–29], we currently need new tools to study such systems. Here, we developed a computational framework of information theory that alleviates several drawbacks of existing tools, primarily by allowing efficient analysis of complex models with multiple inputs and multiple outputs. The method allowed us to provide an insight to one of the long-standing problems in signaling: how type I and type III interferon signaling can be recognized by cells despite activating the same signaling effectors.

## Results

### Quantification of information transfer in signaling systems.
Within information theory, a signaling system is typically considered as a probability distribution $P(Y|X = x)$ that for a given level of input, $x$, elicits output, $Y$. In a typical example, the input is the concentration of a ligand that activates a receptor, and the output is the activity of a signaling effector, which might be the nuclear concentration of an activated transcription factor. The output, $Y$, carries information about the level of the input, $x$. How much information is transferred depends on the signaling system itself, i.e., on its noise levels and sensitivity to changes of input values, as well as on how frequently different input values are transmitted. To illustrate the latter, consider two possible sets of input values. One set of input values generates similar and/or irreproducible outputs, while the other generates distinct and reproducible outputs. If a pathway encounters signals from the first set more frequently than from the second one, its information transfer will be on average lower. The mutual information, $I(X,Y)$, quantifies information transfer of a given signaling system, $P(Y|X = x)$, that encounters input values following a given distribution, $P(X)$ (see Methods). The maximal mutual information, with respect to all input distributions, termed information capacity, $C^*$, quantifies information transfer under the most favorable distribution of input values

$$C^* = \max_{P(X)} I(X, Y). \tag{1}$$

The distribution for which the maximum of mutual information is achieved is called the optimal input distribution and denoted as $P^*(X)$. The information capacity, $C^*$, is expressed in bits, and $2^{C^*}$ can be interpreted, within the Shannon's coding theorem[17,30,31], as the number of input values that the system can effectively resolve based on the information contained in the output. For instance, if $C^* = 2$, there exist four concentrations that can be distinguished with, on average, negligible error. Available approaches to compute information capacity are briefly described in Methods, whereas more background on information theory is provided in Section 1 of Supplementary information (SI).

### Efficient calculation of information capacity in complex models.
In a general setting, calculation of the information capacity, $C^*$, is computationally expensive, if not prohibitive. Here, we propose a framework to study information flow in biochemical signaling models that alleviate several of the important drawbacks related to available approaches[19,22,25,32]. Specifically, the proposed framework is based on analytical solutions. This, in turn, leads to an efficient computational algorithm that accounts for the complexity of signaling, most importantly multidimensional inputs and outputs.

We propose to calculate the information capacity, using an asymptotic approach. Precisely, we consider a system with an output, $Y_N = (Y^{(1)},...,Y^{(N)})$, that consists of $N$ independent copies of $Y \sim P(\cdot|X = x)$, where $Y$ itself can be multidimensional, e.g., a time series of induced levels of transcription factors. Biologically, $N$ can be interpreted as the number of cells that independently sense the signal, $X$. The corresponding information capacity

problem is then written as

$$C_N^* = \max_{P_N(X)} I(X, Y_N). \tag{2}$$

$C_N^*$ quantifies information about the input, $X$, jointly stored in $N$ cells. For large $N$, Eq. (2) has an exact and computationally efficient solution based on the Fisher information matrix (FIM),

$$\text{FIM}_{ij}(x) = \mathbb{E}\left[\frac{\partial \log P(Y|X=x)}{\partial x_i}\frac{\partial \log P(Y|X=x)}{\partial x_j}\right], \tag{3}$$

where $i$ and $j$ refer to elements of the vector $x = (x_1,...,x_k)$, i.e., multidimensional input. Specifically, it has been shown in the statistical theory of reference priors[30,33] that if FIM is non-singular, i.e., all inputs have a non-redundant impact on the output, then

$$P_N^*(x) \xrightarrow[N\to\infty]{} P_{JP}^*(x), \tag{4}$$

where

$$P_{JP}^*(x) \propto \sqrt{|\text{FIM}(x)|}, \tag{5}$$

and $|\cdot|$ denotes the matrix determinant. The distribution $P_{JP}^*(x)$ is known as the Jeffrey prior (JP). Similarly, it can be shown, see Section 1.4 SI and ref. [30], that

$$C_N^* - \frac{k}{2}\log_2(N) \xrightarrow[N\to\infty]{} C_A^*, \tag{6}$$

where $k$ is the dimension of input, and

$$C_A^* = \log_2\left((2\pi e)^{-\frac{k}{2}}\int_{\mathscr{X}}\sqrt{|\text{FIM}(x)|}\,dx\right), \tag{7}$$

where $\mathscr{X}$ is the space of signal values, $x$.

As suggested by Eq. (6), we will call $C_A^*$ as the asymptotic information capacity, where asymptotics is meant with respect to the number of cells, $N$. Equation (6) implies that $C_A^*$ can be used to approximate the joint capacity of $N$ cells

$$C_N^* \approx C_A^* + \frac{k}{2}\log_2(N). \tag{8}$$

The approximation demonstrates that the joint capacity of $N$ cells depends on the baseline, asymptotic, capacity, $C_A^*$, and on the number of cells via $\frac{k}{2}\log_2(N)$, where the latter term vanishes for $N = 1$. Therefore, in terms of Eq. (8), the asymptotic capacity $C_A^*$ can be interpreted as the contribution of an individual cell to the capacity of an ensemble of $N$ cells. Equivalently, the number of inputs resolvable by $N$ cells increases linearly with $N^{\frac{k}{2}}$ at the rate $2^{C_A^*}$

$$2^{C_N^*} \approx 2^{C_A^*} \cdot N^{\frac{k}{2}}. \tag{9}$$

In terms of Eq. (9), the asymptotic capacity $C_A^*$ defines a rate, at which the number of resolvable states increases with $N$.

Importantly, asymptotic capacity, $C_A^*$, can take negative values, which has a precise interpretation. The scaling law of Eqs. (8) and (9), which is warranted to be correct by convergence in Eq. (6), implies that $C_A^*$ must be allowed to take negative values. If $C_A^*$ was guaranteed to be positive then, any signaling system composed of $N$ cells would be guaranteed to have the capacity $C_N^*$ larger than $\frac{k}{2}\log_2(N)$, which obviously is not the case. In other words, if the number of resolvable inputs $2^{C_N^*}$ increases slowly with $N$ then $2^{C_A^*}$

must be accordingly small, which means negative $C_A^*$. For illustration, consider two systems with asymptotic capacities, $C_A^*$, of, say, $-1$ bit and 1 bit. Then, the capacity $C_N^*$, of the first is smaller by 2 bits compared to the second, for large $N$. Equivalently, the number of resolvable inputs of the first systems increases at the fourth of the rate of the latter. Besides, Eq. (7), implies that for systems with low Fisher information the number of resolvable inputs increases slowly with $N$.

Conveniently, $C_A^*$ reduces the problem of calculating the information capacity to the problem of calculating the FIM and we propose to take advantage of this. Fisher information can be calculated for systems with multiple inputs and outputs, and therefore the above approach allows simple computation of information capacity for such systems. To the best of our knowledge, this method has not been used to analyze biochemical signaling, most likely due to technical difficulties in calculating the FIM, which was, to a considerable degree, alleviated by methods recently developed[34,35]. In Section 6 of SI we discuss in details how FIM can be calculated in different scenarios.

**Asymptotic capacity does not deviate substantially from non-asymptotic capacity in the test model**. The asymptotic capacity, $C_A^*$, and the capacity of an individual cell, $C_1^*$, are related but not the same quantities. As we discuss in Section 1 of SI, differences arise from non-identical optimal input distributions of single cells and population of cells as well as the way in which information from different cells adds up. In the literature, so far, the interest in $C_1^*$ is dominating. Therefore, even though $C_A^*$ has a meaningful interpretation on its own, we have compared values of $C_A^*$ and $C_1^*$ in a test model. Blahut–Arimoto algorithm was used to calculate the exact $C_1^*$. In the comparison we have also included the established, and virtually the only available method to approximate $C_1^*$, i.e., the small noise approximation[19], denoted here as $C_{SN}^*$. We have designed a test model, for which all methods are computationally feasible, and which challenges the assumption of our method, i.e., asymptotics, and of the small noise approximation, i.e., limited stochasticity. Precisely, we considered a model of a biochemical sensor described by the binomial distribution $Y \sim \text{Bin}(h(S), L)$ with the output $Y$ being the number of active sensors and $L$ being the copy number of sensors. The probability of the sensor being active was assumed to be the Michaelis–Menten function, $h(S) = S/H/(1 + S/H)$, with $S = X + X_F/\lambda$, where $X$ is the concentration of a cognate and $X_F$ of a non-cognate ligand, and $\lambda$ is the selectivity factor (the ratio of the binding affinities, $K_d$'s, of the non-cognate and cognate ligands, $\lambda = \frac{H_F}{H}$). The higher the value of $\lambda$, the less likely the receptor binds the false ligand. We have assumed that the concentration of the true ligand, $X$, is the input of the system and varies according to the optimal input distribution, $P^*(X)$, whereas the variability of the non-cognate ligand, modeled as the probability distribution $P(X_F)$, is the source of noise that leads to information loss. For calculation of $C_1^*$ with Blahut–Arimoto algorithm, we used a complete model without any approximations.

Changing the settings of this model allowed us to challenge the tested methods thoroughly. In total, we have considered 27 different scenarios by combining different variants of the probability distributions $P(X_F)$; sensor copy number, $L$; and of the selectivity factor, $\lambda$. In each scenario, we have calculated capacities as a function of the standard deviation of the distribution $P(X_F)$, denoted as $\sigma_{X_F}$. Relative deviations of $C_A^*$ and $C_{SN}^*$ from $C_1^*$, averaged over all considered scenarios of the test model, are presented in Fig. 1, whereas comparison for each scenario is presented in Supplementary Figures 1–3. For limited variability, i.e., small $\sigma_{X_F}$, both methods have similar accuracy. When the variability increases both methods become less accurate;

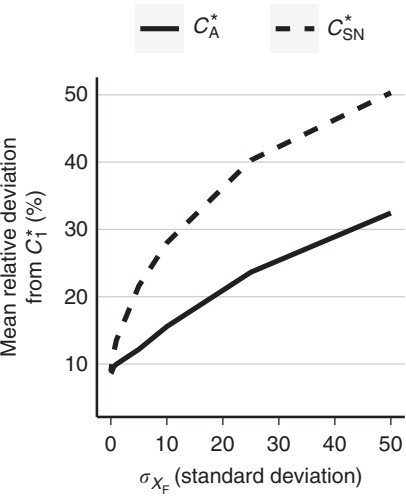

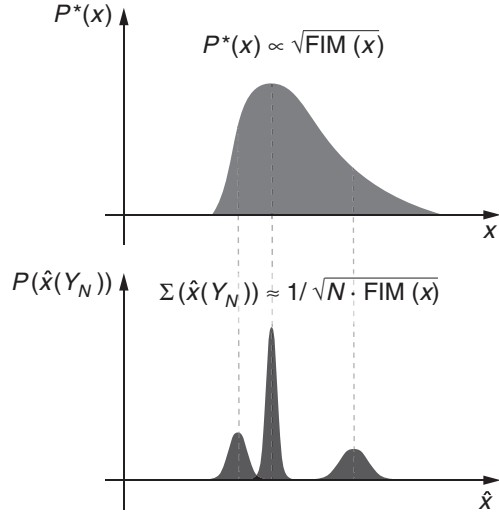

**Fig. 1** Asymptotic capacity, $C_A^*$, does not deviate substantially from non-asymptotic capacity, $C_1^*$, in the test model. Average deviation of $C_A^*$ and $C_{SN}^*$ from $C_1^*$ calculated based on all 27 considered scenarios of the test model are plotted against the standard deviation of the model noise, $\sigma_{X_F}$. $C_1^*$ was computed without approximations using BA algorithm. Scenarios used to calculate the deviations were combinatorially generated by considering different values of $L$, $\lambda$, and non-cognate ligand distributions (see Section 2 of SI). Capacities for each considered scenario are shown in Supplementary Figure 1

**Fig. 2** Information transmission is maximized when frequent signals are recognized with high precision. Given the asymptotic interpretation (10), for a system with one-dimensional input, the standard deviation of the signal estimate is given as $\Sigma(\hat{x}) \approx 1/\sqrt{N \cdot \mathrm{FIM}(x)}$. Then, the asymptotically optimal input distribution, $P_{JP}^*(x)$, is defined in terms of uncertainty of inferences, $P_{JP}^*(x) \propto 1/\Sigma(\hat{x}(Y_N))$. In the optimal scenario, signals occur with frequencies inversely proportional to the uncertainties

however, $C_A^*$ has half lower error compared to $C_{SN}^*$. High variability violates the assumption of the small noise approximation, which explains higher error for high $\sigma_{X_F}$. Lower approximation accuracy of $C_A^*$ results from the lack of asymptotics, i.e., $N = 1$. When using $C_A^*$ or $C_{SN}^*$ as approximations of $C_1^*$, which is a positive quantity, one should monitor for negative values and set approximation to zero. Supplementary Figure 1 shows that in the test model both approximations fell below zero for several model settings, specifically these corresponding to low copy number and highest considered $\sigma_{X_F}$. In Section 1.5 of SI we present an auxiliary approximation of $C_1^*$ that is guaranteed to be positive, but is computationally not as efficient as $C_A^*$.

In summary, our numerical analysis demonstrates that $C_A^*$ provided a more accurate approximation of $C_1^*$ than $C_{SN}^*$. The approximation error is at the order of maximum 30%, which indicates that the asymptotic capacity, $C_A^*$, served as a reliable approximation of the capacity of an individual cell, $C_1^*$.

**Information transmission is maximized when frequent signals are recognized with high precision**. How much information is transferred in a given signaling system depends on three factors: (i) sensitivity of the output to changes in the input, (ii) variability of output given input, and (iii) how frequently do different inputs occur. The first two are modeled by the input–output distribution, $P(Y|X = x)$, and the third is represented by the maximization problem in Eq. (2). Here we show that our approach allows for an insightful interpretation of the input distribution that is optimal for signaling. Precisely, consider an asymptotically efficient estimator, $\hat{x}(Y_N)$ value, $x$, i.e., an estimator that achieves lowest possible variance for large data, e.g., maximum likelihood estimator. Then, the variance of this estimator, $\Sigma(\hat{x}(Y_N))$, is asymptotically described by the inverse of the Fisher information

$$\Sigma(\hat{x}(Y_N)) \xrightarrow{N \to \infty} (N \cdot \mathrm{FIM}(x))^{-1}. \tag{10}$$

Given the above, the optimal distribution of inputs, $P_{JP}^*(x) \propto \sqrt{|\mathrm{FIM}(x)|}$, is defined in terms of the uncertainty of

inferences, $\Sigma(\hat{x}(Y_N))$, that cells can draw about the input value, $x$. Precisely, for large $N$, $P_{JP}^*(x) \propto 1/\sqrt{|\Sigma(\hat{x}(Y_N))|}$. Therefore, the optimal distribution, $P_{JP}^*(x)$, states that the system performs best in terms of the information capacity if frequent values are recognized and processed with high precision, whereas more rarely occurring signals need not be transmitted with similarly high accuracy. This is visualized in Fig. 2 for a scenario with a one-dimensional input: in the optimal scenario signals occur at a frequency that is proportional to the inverse of the uncertainty measured as the standard deviation of the estimate of the signal, $\sqrt{\Sigma(\hat{x}(Y_N))}$.

Signaling precision is also closely related to the discrimination error. This is relevant as the information capacity per se does not indicate which exact states can be effectively discriminated. Precisely, consider two close input values $x_0$ and $x_1$, and the probability, $\varepsilon(x_0, x_1, Y_N)$, of not detecting the change $x_0 \to x_1$ based on observations $Y_N$. Within the statistical framework of hypothesis testing, the probability $\varepsilon(x_0, x_1, Y_N)$ is approximated as[17]

$$\varepsilon(x_0, x_1, Y_N) \approx e^{-N(x_1-x_0)\mathrm{FIM}(x_0)(x_1-x_0)^T}. \tag{11}$$

Therefore, changes in input concentrations that are easily recognized are these along sensitive directions of the FIMs. These directions and can be determined in our framework.

**Signaling dynamics allows discrimination between identity and quantity of type I and type III interferons**. In order to demonstrate how our method can be applied to provide a unique insight regarding the functioning of signaling pathways, we have addressed the problem of the type I and type III interferons signaling. Both IFN types induce the same signaling effectors and it is currently not clear how their identity and quantity is recognized by cells to induce distinct physiological responses[12–15,36]. Both IFN types have several variants and here we have selected IFN-$\alpha$ and IFN-$\lambda$1 as representatives of type I and type

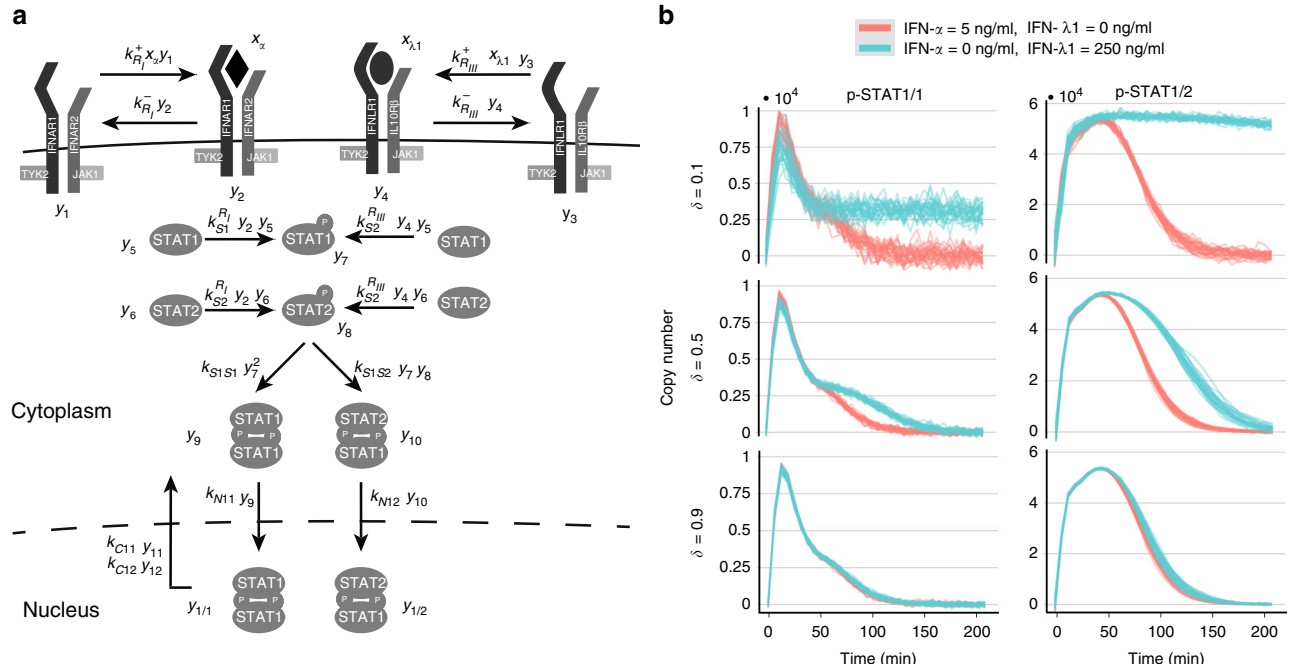

**Fig. 3** IFN-$\alpha$ and IFN-$\lambda$1 activate the same signaling effectors but induce different signaling dynamics. **a** Model representation of the type I and type III IFNs signaling used to construct the input–output probability distribution, $P(Y|X)$. IFN-$\alpha$ exerts its action through cognate two subunits receptor complex IFNAR1/IFNAR2, whereas IFN-$\lambda$1 signals through two subunit receptor complex IFNLR1/IL10R$\alpha$. Both of receptor complexes are pre-associated with JAK1 and TYK2 tyrosine kinases[50, 51]. Simplistically, receptor ligand binding results in tyrosine phosphorylation of STAT1 and STAT2 proteins. These, denoted as p-STAT1 and p-STAT2, respectively, form p-STAT1/1 homodimers and p-STAT1/2 heterodimers and translocate to the nucleus. Dephosphorylation in the nucleus results in nuclear export of STATs and makes them available to subsequent phosphorylation/dephosphorylation cycles. **b** Samples of the model output $Y$ (copy numbers of nuclear p-STAT1/1 and p-STAT1/2 dimers) in response to 30 min stimulation with IFN-$\alpha$ (green; 5 ng/ml) or IFN-$\lambda$ (red; 250 ng/ml) for three levels of $\delta$ and $c_v = 0$. In each panel 30 trajectories are plotted. Parameters used to simulate the mode are given in Supplementary Tables 1 and 2

III IFNs, respectively. IFN-$\alpha$ exerts its action through cognate two subunits receptor complex IFNAR1/IFNAR2, whereas IFN-$\lambda$1 signals through two subunit receptor complex IFNLR1/IL10R$\alpha$. Simplistically, receptor ligand binding induces a cascade of events. The cascade culminates with phosphoryled forms of STAT1 and STAT2 proteins translocating to the nucleus as homodimers (p-STAT1/1) and heterodimers (p-STAT1/2), where they bind DNA to specific cognate sites (Fig. 3a). The mechanism that explains the differential physiological effect of IFN-$\alpha$ and IFN-$\lambda$1 despite inducing the same signaling effectors is largely unknown[12–14]. Recent data[12–14,37], however, support the hypothesis that a differential temporal profile, understood as time series of the copy numbers of nuclear p-STAT1/1 homodimers and p-STAT1/2 heterodimers, carries information about identity and quantity of both IFNs and is further propagated by the gene expression machinery into distinct physiological responses. For instance, western blot experiments show a prolonged phosphorylation in response to IFN-$\lambda$1 compared to IFN-$\alpha$[14].

Our framework provides a natural, and computationally feasible, framework to address IFN discrimination problem. As four resolvable states are required to distinguish between presence and absence of two stimuli, the capacity $C_N^* \geq 2$ can be interpreted as the potential of a population of $N$ cells to distinguish both identity and quantity of the two IFNs. Moreover, Eqs. (8) and (11) imply that if FIM is non-singular the capacity $C_N^*$ can be arbitrarily increased and the discrimination error $\varepsilon(x_0, x_1, Y_N)$ arbitrarily decreased with the population size $N$. Therefore, information capacity and FIMs constitute suitable tools to determine how information about identity and quantity of both IFN is encoded in signaling responses.

To this end, we have built a probabilistic model of the pathway's input–output relationship, $P(Y|X = x)$. Construction of the model was accomplished by assembling and refining model components of the JAK-STAT signaling available in literature[38–40] (Fig. 3a and Section 3 of SI). The input $x = (x_\alpha, x_{\lambda 1})$ consists of concentrations of IFN-$\alpha$ and IFN-$\lambda$1, respectively. We assumed that the pathway is exposed for 30 min to stimulation with a mixture of IFNs specified by the input. The output is defined as $Y = (Y_{1/2}(t_1), Y_{1/1}(t_1),...,Y_{1/2}(t_n), Y_{1/1}(t_n))$, where $Y_{1/2}(t_i)$ and $Y_{1/1}(t_i)$ denote copy numbers of nuclear of p-STAT1/2 heterodimers and p-STAT1/1 homodimers, respectively, at time $t_i$. Times $t_1,...,t_n$ serve as a proxy of the complete temporal profile. To account for signaling noise, we assumed that the stochasticity results from: (i) randomness of individual reactions and (ii) also cell-to-cell variability in the copy numbers of STAT1 and STAT2 molecules as well as type I, $R_I$, and type III, $R_{III}$, receptor complexes. The two noise sources are seen as main contributors of cell-to-cell heterogeneity in general[41] and IFN signaling, specifically[42]. The copy numbers of the above entities per cell was assumed variable with the same coefficient of variation

$$c_v = \frac{\sigma_i}{\mu_i}, \qquad (12)$$

where $\mu_i$ is the mean copy number per cell, and $\sigma_i$ is its standard deviation, for $i \in \{STAT1, STAT2, R_I, R_{III}\}$. Further, we considered coefficient of variation from 0.3 to 1.5 to reflect typically measured values[43]. Importantly, the model is in line with the present biochemical knowledge[14,37] by allowing the only difference in responses to arise from the different kinetics of

both receptor complexes. We quantified the difference in receptor kinetics using the ratio of deactivation rates of the type III and type I receptor complexes, $k_{\bar{R}_{III}}$ and $k_{\bar{R}_I}$, respectively (see Sections 3.2–3.3 of SI),

$$\delta = \frac{k_{\bar{R}_{III}}}{k_{\bar{R}_I}}, \qquad (13)$$

which we call the differential kinetics coefficient. For a given value of $\delta$ (e.g. 0.5), upon activation, the type I receptor complex remains active on average $1/\delta$ (e.g. 2) times shorter than the type III receptor. As responses to IFN-$\lambda 1$ are prolonged compared to IFN-$\alpha$[14], we have considered $\delta \in (0, 1)$. Values close to 0 and 1 denote dissimilar and similar receptor kinetics, respectively. The model was numerically simulated within the framework of the linear noise approximation that allows efficient calculation of the FIMs[34,35] using literature values of kinetic parameters (see Supplementary Table 2). As shown in Fig. 3b the model provides responses qualitatively consistent with experiments that show prolonged responses to IFN-$\lambda 1$ compared to IFN-$\alpha$[14]. The strength of this effect is controlled by the parameter $\delta$. Low $\delta$ implies a significantly longer response to IFN-$\lambda 1$, whereas for $\delta$ close to 1 responses to both IFNs appear to be indistinguishable.

First, we examined the potential of the differential signaling dynamics to discriminate between identity and quantity of IFNs under noise limited to stochasticity of individual reactions, $c_v = 0$. To this end, we considered outputs, $Y$, with different end time points, $t_n$'s, so that they contain only the information available to the cell until time $t_n$. For the values of $\delta$ used in Fig. 3b, we plotted the information capacity, $C_A^*$, as a function of $t_n$ (Fig. 4a) as well as corresponding representative FIMs (Fig. 4b). For early times, the capacities, $C_A^*$ are below 0, further, with increasing $t_n$, raise over 2 bits, and finally plateau. The time-windows of rapid increase coincide with times, at which stimulation with different combinations of the two IFNs generates distinguishable output trajectories (compare with Fig. 3b and Supplementary Figures 4 and 5). Correspondingly, FIM is singular only for early $t_n$ and high $\delta$. Also, it becomes close to orthogonal for late $t_n$ and small $\delta$.

These results indicate that for limited signaling noise, differential signaling dynamics has a potential to ensure discriminability between the two IFNs. Precisely, for all $\delta$'s and late $t_n$, $C_A^*$ reaches high values, and FIMs are non-singular. Therefore, at the population level, both capacity, $C_N^*$, arbitrarily increases (Eq. 8), and the discrimination error arbitrarily decreases (Eq. 11), with $N$. Moreover, using $C_A^*$ as an approximation of $C_1^*$, which can be safely done for high values of $C_A^*$, we can also conclude that the differential signaling dynamics results with the single-cell capacity, $C_1^*$, significantly higher than 2 bits. Two bits is a minimum necessary condition to encode four input values, e.g., presence and absence of two stimuli. However, Shannon information alone does not tell us which exact input values can be discriminated. Therefore, we can conclude only that individual cells can resolve at least four combinations of both IFN concentrations.

**Population level discrimination is possible even at high noise and with minor kinetic differences**. The above analysis demonstrated that with limited noise signaling dynamics ensures discrimination between both IFNs. Interestingly, the discrimination is possible even with modest differences in the kinetics of both receptors, i.e., $\delta = 0.9$, which corresponds to 10% difference in the receptors deactivation rates. Noise in signaling processes is, however, not limited to stochasticity of signaling reactions. In mammalian signaling, the noise is thought to be dominated by the copy number variability of signaling components[8]. Therefore, we have considered several noise levels, and examined how the

information content of the complete temporal profile, $t_n = 180$, depends on the values of the differential kinetics coefficient, $\delta$. Fig. 4c presents the capacity, $C_A^*$, as a function of $\delta$ for a range of biologically feasible[43] values of $c_v$. Fig. 4d shows corresponding FIMs. Not surprisingly, both noise and lack of kinetic differences can severely compromise the information transfer (Fig. 4c). $C_A^*$ falls substantially below 2 bits, reaching negative values for high noise and similar kinetics. On the other hand, representative FIMs are non-singular for all noise levels and values of $\delta$.

The above results primarily show that discriminability at the population level can be achieved even with minor differences in kinetic rates, and despite high noise levels. This is implied by Eqs. (8) and (11). As indicated by Eq. (8), high population capacity, $C_N^*$, can be ensured by large $N$, as long as $C_A^*$ is not prohibitively low. Similarly, Eq. (11) shows that low discrimination error, $\varepsilon(x_0, x_1, Y_N)$, can be ensured by large $N$, as long as FIMs are non-singular. Both conditions are satisfied in the considered scenarios as shown in Fig. 4c, d. In addition, negative values of $C_A^*$, for high $\delta$ and $c_v$, indicate slow increase of the overall number of resolvable input values with $N$ (Eq. 9).

Moreover, our analysis demonstrates that discriminability at the population level does not require discriminability at the single-cell level. This conclusion can be made on the following ground. The capacity of two bits is a necessary condition to encode four input values, e.g., presence or absence of both IFNs. In other words, a system with capacity lower than two bits does not have a sufficient discriminatory power to resolve presence and absence of the two IFNs. Therefore if $C_1^* < 2$ the discrimination at the single-cell level is not possible. Here, we calculated $C_A^*$ not $C_1^*$, which can only serve as an approximation of $C_1^*$. However, $C_A^*$ falls substantially below two bits. Therefore, even if $C_A^*$ was not a very accurate of approximation of $C_1^*$, low values of $C_A^*$ strongly indicate that $C_1^*$ is smaller than two bits, which demonstrates that there is no discriminability at the single-cell level. On the other hand, as argued in the previous paragraph, discriminability at the population can be achieved by increasing $N$.

Interestingly, our analysis also highlights the role of kinetic rates in efficient information transfer. Primarily, the model predicts that at the population level, the discriminability between the two IFNs can be achieved even at high noise with differences in kinetic rates at the order of 10%. This suggest that even minor divergence of evolutionary related receptors might suffice to augment information transfer. This prediction is in line with the highly cross-wired architecture of signaling pathways[29,44]. Secondly, Fig. 4c shows that loss of information due to noise can be compensated by stronger kinetic differences, and vice versa. This trade-off emphasizes the divergence of kinetic rates as an easily accessible evolutionary strategy of increasing information transfer. Reduction of noise level requires an increase in the number of signaling molecules or/and sophisticated control mechanisms. On the other hand, alteration of receptor kinetic rates can be caused by a single mutation[45].

Overall, our model predicts that the population can correctly decode information even in cases where single cells cannot, due to high noise or similar receptor kinetics. The question, however, arises how the population should be able to make correct decisions based on low capacity in single cells. To illustrate this, consider the expression of IFNs induced genes as a downstream output. Both IFNs induce expression of hundreds of gene, including several chemokines from CXCL and CCL family[12,46,47]. Specifically, it has been shown that temporal profiles of CXCL10 expression differ in response to both considered IFNs[12].

One of the main function of these chemokines is to attract different types of leukocytes to a site of an infection. Therefore, concentration and timing of these chemoattractants can be seen

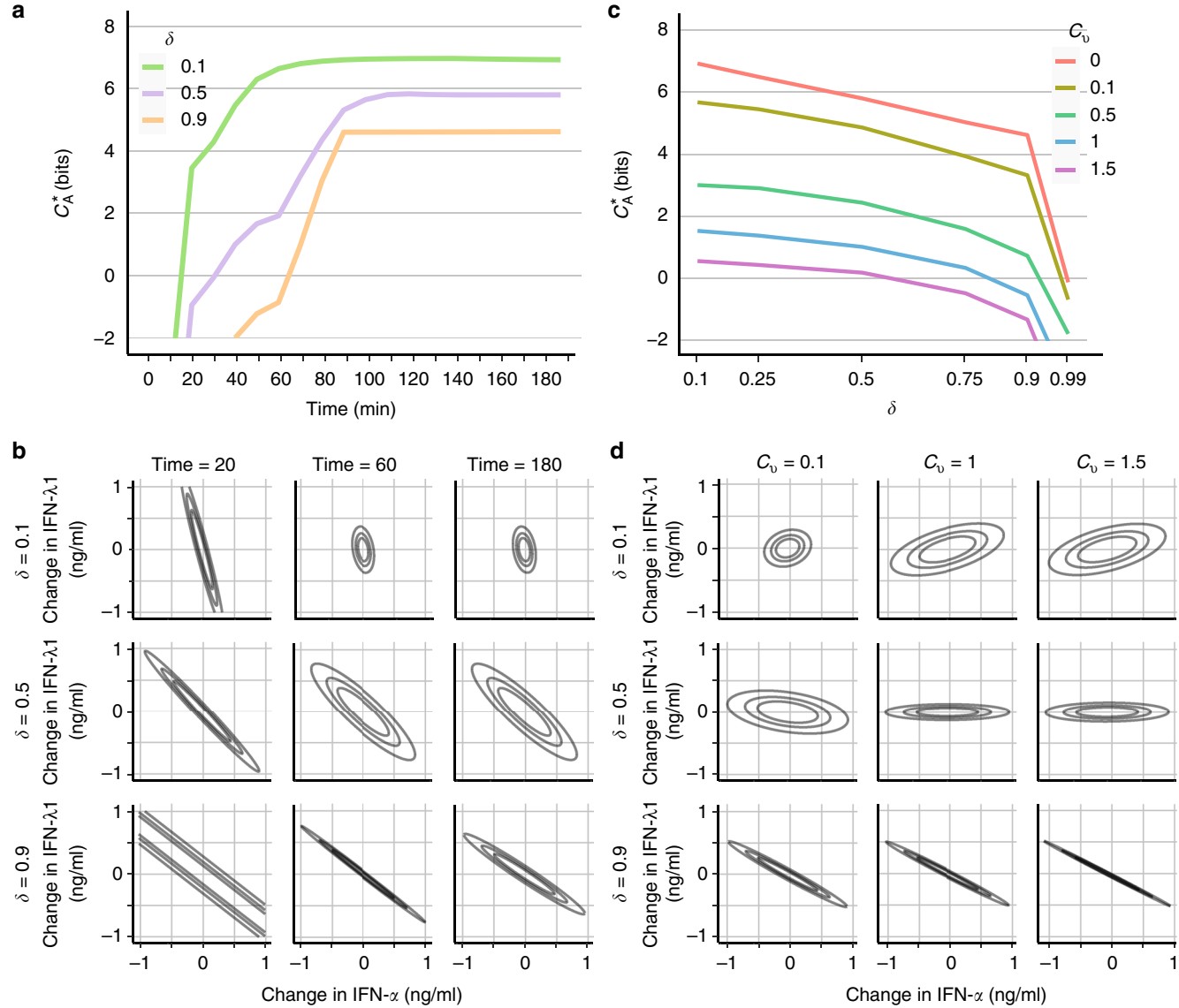

**Fig. 4** Signaling dynamics has a potential to transfer information about identity and quantity of IFN-$\alpha$ and IFN-$\lambda$1. **a** Information capacity, $C_A^*$, as a function of the length of the output, $t_n$, for different values of the differential kinetics coefficient, $\delta$, and the coefficient of variation $c_v = 0$. **b** Isolines of the quadratic forms $(x - x_0)^T \text{FIM}(x_0)(x - x_0)$ for FIMs corresponding to different values of $t_n$ and $\delta$. Given Eq. (11), these isolines show concentration changes that are detected with the same error. Further isolines have smaller error. Singularity (parallel lines) implies lack of discriminability. **c** Information capacity, $C_A^*$, as a function of the differential kinetics coefficient, $\delta$ for different values of $c_v$ assuming maximal considered output length, i.e., $t_n = 180$. **d** As in **b** but for FIMs corresponding to **c**. Non-singularity of FIM indicates that both IFNs can be discriminated for sufficiently large number of cells. Modeling details: In **b** and **d**, we used $x_0$ given by the receptors $K_d$'s, i.e., $x_0 = (k_-^I/k_+^I, k_-^{III}/k_+^{III})$. Remaining parameters used for computations are given in Supplementary Tables 1 and 2

as a decision of cellular population regarding which and how many leukocytes are needed at a given time. Concentration and timing are controlled jointly by a large number of cells due to averaging of secretions in the intercellular space. In consequence, the chemokine concentration depends on the information encoded in nuclear levels of the p-STAT1/1 and p-STAT1/2 dimers in multiple cells. Therefore, even if the capacity of individual cells is low, and as a result expression of the chemokine can be only vaguely controlled by IFNs level, the high joint capacity of $N$ cells may lead to a finely tuned level of the chemokine in the intercellular space, as the differences in secretion of individual cells would average out. To further hypothesize how the high capacity of a population and low individual capacity may be utilized, one can also anticipate a different scenario. An initial stimulus leads to subsequent

rounds of cell-to-cell communication through paracrine signaling. Low individual capacity implies that initially the stimulus is recognized with low precision. In subsequent rounds of communication information is exchanged between cells, and as a result, the initial stimulus may lead to finely tuned responses at later times.

Although the above hypothetical mechanisms are in line with current understanding of IFN signaling, they imply the need for more detailed experimental testing. They also rise an essential questions regarding signaling processes: does effective information transfer require discriminability of IFNs, and signaling ligands more generally, at the single-cell level, or population level suffices? So far, differential IFNs signaling dynamics have been observed at the population level[12–15,36]. Experimental confirmation of our theoretical prediction would be of high relevance for

reconciling the single-cell stochasticity with fine-tuned tissue level responses.

## Discussion

Information theory appears to have a potential to promote further understanding of how cells translate information about complex stimuli into distinct activities of the pathway's effectors using pleiotropic and stochastic mechanisms. Our theoretical methodology establishes a general and computationally efficient framework that enables analysis of models with multiple inputs and outputs. Importantly, it also accounts for the temporal aspect of signaling. Here, we have shown that even in the presence of significant noise information about identity and quantity of IFN-$\alpha$ and IFN-$\lambda$1 can be transmitted despite shared network components. Discriminability at the population level can be achieved without discriminability at the single-cell level, and with only small differences in receptor kinetic rates. So far, signaling responses to both IFNs have been measured at the population level only[12–15,36]. Our analysis suggests that further verification at the single-cell level could provide interesting conclusions regarding how information processing differs between single cells and cellular populations. Scenarios of pleiotropic signaling, similar to IFNs, are common in signaling, e.g., Wnt, BMP[29], as well as in GPCR signaling[48,49]. Therefore, our framework seems to offer an attractive opportunity to gain further insight into the functioning of many complex signaling systems.

## Methods

**Mutual information**. Within information theory, quantification of information transfer of a given signaling system, $P(Y|X = x)$, is performed in reference to the distribution of input values $P(X)$. Although, randomness of output, $Y$, prevents the system from resolving a precise value of the input, $x$, the uncertainty regarding input values cannot be higher than the uncertainty associated with the input distribution, $P(X)$. Uncertainty is usually quantified by entropy

$$H(X) = -\int_{\mathscr{X}} \log_2(P(x))P(x)\mathrm{d}x, \tag{14}$$

where $\mathscr{X}$ is the space of possible values of the signal, $X$.

Observation of output has a potential to reduce uncertainty regarding input value. Via the Bayes formula, plausible inputs that generated a specific output value, $y$, are represented as the probability distribution $P(X|Y = y) = \frac{P(Y=y|X)P(X)}{P(Y=y)}$. Uncertainty regarding input value can be then quantified by the entropy of the distribution $P(X|Y = y)$

$$H(X|Y = y) = \int_{\mathscr{X}} \log_2(P(x|Y = y))P(x|Y = y)\mathrm{d}x. \tag{15}$$

As the output is random, averaging $H(X|Y = y)$ over all possible outputs quantifies the average uncertainty regarding the input, given the output, $H(X|Y)$

$$H(X|Y) = -\int_{\mathscr{Y}} H(X|Y = y)P(y)\mathrm{d}y, \tag{16}$$

where $\mathscr{Y}$ is the space of possible values of the output, $Y$. The difference between $H(X)$ and $H(X|Y)$ measures the average reduction in uncertainty regarding the input resulting from observing an output and is referred to as mutual information, $I(X, Y)$, between the input and the output

$$I(X, Y) = H(X) - H(X|Y). \tag{17}$$

More background on information theory is provided in Section 1 of Supplementary methods.

**Existing methods to compute information capacity**. Three main approaches are available to calculate $C^*$ and $P^*(X)$. The state-of-the-art Blahut–Arimoto algorithm is based on convex optimization[25,32] and for systems with continuous variables it requires discretization of input and output values[18]. Although it works efficiently for systems with one-dimensional input and output, optimization may become computationally prohibitive for higher dimensionalities. An alternative approach is offered by the small noise (SN) approximation method[19], which offers an analytical solution, and therefore avoids heavy computations. However, it assumes a limited

stochasticity within the analyzed system. Recently, a method based on density approximation was proposed in ref. [22] to account for temporally resolved outputs.

**Code availability**. Computer code used to generate reported results is available from authors upon request.

## Data availability

The study did not involve any datasets.

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

## Acknowledgements

T.J. was supported by his own funds and the European Commission Research Executive Agency under grant CIG PCIG12-GA-2012-334298, M.K. and K.N. by the Polish National Science Centre under grant 2015/17/B/NZ2/03692. We thank Stefan Grünert, Marek Kochańczyk, Margaritis Voliotis, and Christopher Zechner for their helpful comments during the preparation of this manuscript. The model of bio-chemical sensor exposed to non-cognate ligand was inspired by discussions with Prof. Dan S. Tawfik.

## Author contributions

T.J. and M.K. designed research; T.J., K.N., and M.K. performed research; T.J., K.N., S.F., M.P.H.S and M.K. analyzed data; and T.J., K.N., M.P.H.S and M.K. wrote the paper.

## Additional information

**Competing interests:** The authors declare no competing interests.

