## [Peer Review File · Nature Communications]

Reviewers' comments:

Reviewer #1 (Remarks to the Author):

The topic of applying information theoretic approaches to signaling is topical and of potentially broad interest.

The present paper is entirely theoretical. Rather than measuring mutual information from experimental data the authors utilize simulated data, from mathematical models of signaling pathways – models that are not experimentally validated, and the specifics of the molecular interactions questionable and incomplete. Further the sources of noise are considered without experimental justification in terms of quality or quantity. As such the study remains entirely theoretical and the conclusions are tentative, conditioned on the various assumption. This is more a computational methods paper than a paper that reports insights about biological regulation. However, the applicability and utility of the method seems marginal, and not of broad interest.

Reviewer #2 (Remarks to the Author):

The manuscript presents a framework for efficient estimation of information capacity in biochemical Signalling systems. In this framework the information capacity of a Signalling pathway at the single cell levels is approximated by a quantity that the authors call asymptotic information capacity. As I understand it, this quantity is obtained by deriving an approximate expression for the information capacity of a large population of cells and regressing this result back to the single cell level. The authors claim that the main advantage of the framework is computational efficiency in particular in the presence of high dimensional input/outputs. Finally, the authors use their proposed methodology to study how type I and type III interferon Signalling is recognised by cells despite activating the same Signalling effectors.

To my knowledge the methodology that the paper presents is novel and will be of great interest to the wider community of computational biologists. However, I feel that there are a few points that might cause confusion and need to be clarified further.

1. I believe it would be useful if the authors briefly discussed the following points in greater details:

i) the difficulties involved in calculating FIM, especially the ones persisting. (In page 4, the authors state: "To the best of our knowledge, this method has not been used to Analyse biochemical signalling, most likely due to technical difficulties in calculating the FIM, which was, to a considerable degree, alleviated by methods only recently developed").

ii) how the algorithm scales with number of inputs/outputs (looking at fig4, I believe the authors already have this information).

This discussion will make their claims about the advantages of the method stronger.

2. In page 5, the authors state: "The capacity $C_{A \rightarrow *}$ can be interpreted as the potential of a population of N cells to distinguish both identity and quantity of the two IFNs";. I am not sure if this is a correct interpretation, the capacity is giving you in broad terms the number of resolvable states, but does not say anything about which states these are. Could the authors please comment on this, as this could change the way they discuss results in Figure 4.

3. In figure 6. the authors state: "This reveals interesting, if not surprising, insight. Discriminability at the population level is achieved even with minor differences in kinetic rates, despite high noise levels. Also, discriminability at the single cell level is not necessary for discriminability at the population level". I'm not sure if this is surprising at all. Isn't it what you intuitively expect: the

more cells you look the more power to resolve inputs. Also, the authors need to keep in mind that they are not calculating 'discriminability' at the single cell level exactly, but approximating it. There needs to be some (small) 'discriminability' at the single cell level for discriminability at the population level (assuming there are not interaction between cells).

4. How should one interpret the fact that the asymptotic capacity could go negative? Since the authors are using it to approximate information capacity (a positive quantity) of single cells, doesn't it make more sense to set it to zero wherever it goes negative?

Reviewer #3 (Remarks to the Author):

The authors present an approach to quantify the information transmitted in signaling pathways. To this end, they make use of a series of well-known concepts such as Shannon's information theory and channel capacity, the Blahut-Arimoto algorithm and the Fisher information matrix. They extend this concept by applying it not just to one cell but to a larger number N of cells. They show that the joint channel capacity for N cells can be expressed by the asymptotic information capacity (using the Fisher information matrix) plus a value proportional to $\log_2(N)$.

They apply this concept to two simple or simplified models, a generic receptor activation model and a model of the JAK-STAT pathway. In these models they vary some of the parameter values, but also the type of input distribution. They simulate the models and obtain values for channel capacities.

The manuscript presents an interesting contribution to the ongoing discussion about stochasticity in signaling systems and how this influences their capacity to precisely interpret input signals in terms of gene regulation as output.

From my point of view, a number of issues remain.

The authors introduce the joint channel capacity for N cells and draw a number of conclusions w.r.t. the response of a cell population instead of single cells. Specifically, they conclude that the population can correctly decode information even in cases where single cells cannot due to high noise or inappropriate receptor kinetics. While the mathematical formalism is straight, it is unclear to me how the population should be able to make correct decisions based on low capacity in single cells if those don't communicate.

Specifically, there is no evidence that this mechanism is realized in cell populations.

For their simple receptor activation model, they use the small noise approximation. Given that this system is small, I wonder why they don't compare to full stochastic simulation, e.g. with Gillespie algorithm as had been done earlier by other authors. That would allow proper assessment of the system also for larger noise as is assumed to be the case in signaling systems.

Again, for this receptor activation model, I don't see why it should be justified to consider that the cognate signal should be without noise. It should come with comparable noise levels as the non-cognate signal. What is presented appears as an unjustified simplification of the model.

Most surprising, the authors present simulations with C_A reaching values smaller than zero. That even comes without any explanation. However, in my humble opinion, this is complete nonsense. Information is by definition non-negative. Hence C_A must also be non-negative. If this cannot be ensured it is not useful as a measure for information capacity.

We thank reviewers for their inspiring comments which, we believe, have led to an improved version of the manuscript. Specifically, our attention was drawn to the need for a more transparent presentation of the introduced information measures. Consequently, we also provided a more explicit interpretation of model's predictions. We believe, we have addressed all specific issues. Our paper, however, remains purely theoretical. Its main contribution is the appropriate analytical and computational framework to describe information flow in noisy and pleiotropic signaling systems. Nevertheless, the framework is deeply rooted in experimental observations and responds to a need of reconciling highly stochastic realm of single cells with finely-tuned tissue and organismal responses. Given that we need new approaches to investigate how living systems gather, process, store and use information, we hope that our theoretical contribution can be appreciated by Reviewer #1.

Main issue

One issue was raised concurrently by two reviewers. Therefore, we give it special attention and discuss it separately, prior to addressing other issues, point by point. Precisely, reviewers 2 and 3 pointed out that **the asymptotic information capacity C^*_A can be negative**, stating what follows.

Rev. #2

4. How should one interpret the fact that the asymptotic capacity could go negative? Since the authors are using it to approximate information capacity (a positive quantity) of single cells, doesn't it make more sense to set it to zero wherever it goes negative?

Rev. #3

Most surprising, the authors present simulations with C^*_A reaching values smaller than zero. That even comes without any explanation. However, in my humble opinion, this is complete nonsense. Information is by definition non-negative. Hence C^*_A must also be non-negative. If this cannot be ensured it is not useful as a measure for information capacity.

We agree that this issue should have been thoroughly explained, and necessary details were missing in the main text. Judging by hindsight, in need for clarity and brevity, we had moved too much material to the supplement, leaving too little of the necessary explanations.

The asymptotic information capacity C^*_A can indeed take negative values, this, however, has a meaningful interpretation. Although it may cause some unease, as we explain further, it does not result from a caveat of our approach but from mathematical properties of how information scales with increasing N (i.e. the number of receivers / cells).

There exist many other widely accepted information theoretic measures that can take negative values. Differential entropy is probably the best-known example (see, for instance, *Elements of Information Theory* by Thomas and Cover). Other examples include interaction information (i.e. certain form of conditional mutual information) or even mutual information in specific settings. (e.g. $H(X)-H(X|Y=y)$ can be negative if $H(X|Y=y)$ is not averaged over all possible y 's as in the definition of standard mutual information: $H(X)-H(X|Y)$). Negative values of the interaction information have been called "redundancy" but conflicting definitions also exist in the literature.

We have made alterations in the manuscript that explain why the asymptotic capacity must be allowed to take negative values and how it should be interpreted in such cases. We have also provided a more detailed explanation of the results of the IFNs signaling model.

Specifically,

- at the end of the section *Efficient calculation of information capacity in complex signaling models*, we have explained why C^*_A must be allowed to take negative values and how it should be interpreted in such cases.
- we have substantially rewritten interpretations of the results of IFNs signaling model i.e., section *Population level discrimination is possible even at a high noise and with minor kinetic differences*;
- we have designated a distinct section of the Supplement (i.e. Section 1.5), in which we describe an auxiliary approximation of C^*_1 that is guaranteed to be positive.

For the convenience of the reviewers, below is an expanded explanation analogous to the one added at the end of the section *Efficient calculation of information capacity in complex signaling models in the main paper*.

Eq. 10 of the main paper

$$C^*_N - \frac{k}{2} \log_2(N) \xrightarrow{N \rightarrow \infty} C^*_A, \quad (10)$$

implies that the joint capacity of N cells, C^*_N , depends on the baseline, asymptotic, capacity C^*_A and on the number of cells via $k/2 \log_2(N)$

$$C^*_N \approx C^*_A + \frac{k}{2} \log_2(N). \quad (12)$$

Equivalently, the number of inputs resolvable by N cells increases linearly with $N^{k/2}$ at the rate $2^{C^*_A}$

$$2^{C^*_N} \approx 2^{C^*_A} \cdot N^{\frac{k}{2}}. \quad (13)$$

In terms of equation (12), the asymptotic capacity C^*_A can be interpreted as the contribution of an individual cell to the capacity of an ensemble of N cells, and can be used as an approximation of the joint capacity C^*_N .

On the other hand, in terms of equation (13), the asymptotic capacity C^*_A defines a rate, at which the number of resolvable states increases with N.

The scaling law of Eq. 12 and 13, which is warranted to be correct by convergence in Eq. 10, imply that C^*_A must be allowed to take negative values. If C^*_A was guaranteed to be positive then, any signaling system composed of N cells would be guaranteed to have the capacity C^*_N larger than $k/2 \log_2(N)$, which obviously is not the case. In other words, if the number of resolvable inputs $2^{C^*_N}$ increases slowly with N then $2^{C^*_A}$ must be accordingly small, which means negative C^*_A .

Eq. 12 and 13 also provide a precise interpretation of the negative values, which is perhaps best explained by an example. Consider two systems with asymptotic capacities, C^*_A , of, say, -1 bit and 1 bit, respectively. Then, the capacity C^*_N of the first is smaller by 2 bits compared to the second, for large N. Equivalently, the number of resolvable inputs of the first systems increases with N at a fourth of the rate of the latter.

It is, of course, true that if C^*_A is negative then it is not a good approximation of C^*_1 which is a positive quantity. Therefore, for values of C^*_A close to zero it should be used with caution when aiming to approximate C^*_1 . In our test model, it is however still better in approximating C^*_1 than the small

noise approximation. It is worth mentioning that the small noise approximation can also be negative.

We think that in addition to its rigorous interpretation, the asymptotic capacity, C^*_A , is also useful as an approximation of C^*_1 , especially given its computational efficiency. However, to account for scenarios where higher accuracy is needed, in SI we propose an auxiliary approximation of C^*_1 . We refer to this approximation as C^*_{JP} . C^*_{JP} is guaranteed to be positive. See Section 1.5 of the supplement for details. C^*_{JP} is more accurate in approximating C^*_1 than C^*_A . However, it requires more computation time. Hence, in the paper, we have focused on C^*_A .

In fact, the quantity C^*_{JP} was already described in the supplement of the previous version of the manuscript; we now also mention it in the main paper and point to the supplementary material

Point-by-point response to reviewers

Below we address specific reviewers'.

Reviewers' comments are in black with key issues in red.

Our response is in blue with key sentences in bold.

Reviewer #1 (Remarks to the Author):

The topic of applying information theoretic approaches to signaling is topical and of potentially broad interest. The present paper is entirely theoretical. Rather than measuring mutual information from experimental data the authors utilize simulated data, from mathematical models of signaling pathways

1.1 _____

- models that are not experimentally validated, and the specifics of the molecular interactions questionable and incomplete.

We agree that our models are incomplete, however, we think they are useful. This we rephrase after C.R. Rao: "All models are wrong, some models are useful"

We think that the usefulness of our models results not from encapsulation all molecular interactions but from illustrating how the information could be processed by signaling systems. More specifically, we provide experimentally testable prediction regarding differences in how information could be processed at the single cell and population level.

Moreover, **we use generic models**, that provide behavior qualitatively similar to experimentally observed IFNs responses. **The use of similar models seems to be common in modeling of various pathways:**

[R1-1] Tay, Savaş, Jacob J. Hughey, Timothy K. Lee, Tomasz Lipniacki, Stephen R. Quake, and Markus W. Covert. "Single-cell NF- κ B dynamics reveal digital activation and analogue information processing." *Nature* 466, no. 7303 (2010): 267.

[R1-2] Adlung, Lorenz, Sandip Kar, Marie-Christine Wagner, Bin She, Sajib Chakraborty, Jie Bao, Susen Lattermann et al. "Protein abundance of AKT and ERK pathway components governs cell type-specific regulation of proliferation." *Molecular systems biology* 13, no. 1 (2017): 904.

[R1-3] Otero-Muras, Irene, Pencho Yordanov, and Joerg Stelling. "Chemical Reaction Network Theory elucidates sources of multistability in interferon signaling." *PLoS computational biology* 13, no. 4 (2017): e1005454.

In addition, we believe that the key components of IFNs signaling have been included in our model, as previously described in the following references

[R1-4] Smieja, Jaroslaw, Mohammad Jamaluddin, Allan R. Brasier, and Marek Kimmel. "Model-based analysis of interferon- β induced signaling pathway." *Bioinformatics* 24, no. 20 (2008): 2363-2369.

[R1-5] Vanlier, Joep, Christian A. Tiemann, Peter AJ Hilbers, and Natal AW van Riel. "An integrated strategy for prediction uncertainty analysis." *Bioinformatics* 28, no. 8 (2012): 1130-1135.

[R1-6] Otero-Muras, Irene, Pencho Yordanov, and Joerg Stelling. "Chemical Reaction Network Theory elucidates sources of multistability in interferon signaling." *PLoS computational biology* 13, no. 4 (2017): e1005454.

[R1-7] Olganier, D. & Hiscott, J. Type I and type III interferon-induced immune response: It's a matter of kinetics and magnitude. *Hepatology* 59, 1225–1228 (2014).

[R1-8] Schreiber, Gideon, and Jacob Piehler. "The molecular basis for functional plasticity in type I interferon signaling." *Trends in immunology* 36, no. 3 (2015): 139-149.

Given that we consider only responses to 30 minutes pulses of IFNs till the maximum of 120 minutes the use of the simplistic model seems to be justified. Moreover, given the parsimony principle, a more complex model is not always better. See for instance,

[R1-9] Mattingly, Henry H., Mark K. Transtrum, Michael C. Abbott, and Benjamin B. Machta. "Maximizing the information learned from finite data selects a simple model." *Proceedings of the National Academy of Sciences* 115, no. 8 (2018): 1760-1765.

for an interesting recent perspective.

1.2

Further the **sources of noise are considered without experimental justification** in terms of quality or quantity.

We had not included references that would qualitatively justify the selection of the considered noise sources. We have corrected this in the current version, where references [R1-1, R1-2, R1-3] listed below are included together with an explanation of why these noise sources have been selected.

Our noise model is in line with the currently available experimental evidence. In addition to the intrinsic stochasticity of cellular reactions, copy number variation signaling components, i.e. receptors and proteins, is considered to be a major source of noise in signaling pathways in general, and in interferon signaling, specifically.

The degree of copy number variability can be different from protein to protein and from cell type to cell type. Typically, however, the coefficient of variation is between 0.3 and 1 as measured in [R1-3]. This range was included in scenarios considered in our study.

[R1-4] Symmons, Orsolya, and Arjun Raj. "What's luck got to do with it: single cells, multiple fates, and biological nondeterminism." *Molecular cell* 62, no. 5 (2016): 788-802.

[R1-5] Levin, Doron, Daniel Harari, and Gideon Schreiber. "Stochastic receptor expression determines cell fate upon interferon treatment." *Molecular and cellular biology* 31, no. 16 (2011): 3252-3266.

[R1-6] Bar-Even, Arren, Johan Paulsson, Narendra Maheshri, Miri Carmi, Erin O'Shea, Yitzhak Pilpel, and Naama Barkai. "Noise in protein expression scales with natural protein abundance." *Nature genetics* 38, no. 6 (2006): 636.

1.3

As such **the study remains entirely theoretical** and the conditioned on the various assumption. This is more a computational methods paper than a paper that reports insights about biological regulation. However, the applicability and utility of the method seems marginal, and not of broad interest.

Clearly, our manuscript is a computational methods paper. Its main contribution is, to the best of our knowledge, a first analytical/computational framework to calculate information capacity for systems with multiple inputs and outputs. Moreover, our approach is sufficiently general and we cannot easily imagine that there are signalling systems for which the important conclusions do not apply, or would not be informative. We aim at the calculation of information capacity for models, which is complementary to the calculation of capacity from experimental data, and in our view is of similar importance. Being able to study theoretically how various factors, e.g. kinetic rates and protein copy number variability impacts information capacity, is important; especially since obtaining similar insight using experimental approaches is at best difficult, and in many cases prohibitive in terms of cost and labor. The method provides insight that can be further tested experimentally. **We believe that what we propose will be of interest for a broad community of biologists, computational and beyond. Our view is thankfully shared by reviewers 2 and 3.**

Reviewer #2 (Remarks to the Author):

The manuscript presents a framework for efficient estimation of information capacity in biochemical Signaling systems. In this framework the information capacity of a Signaling pathway at the single cell levels is approximated by a quantity that the authors call asymptotic information capacity. As I understand it, this quantity is obtained by deriving an approximate expression for the information capacity of a large population of cells and regressing this result back to the single cell level. The authors claim that the main advantage of the framework is computational efficiency in particular in the presence of high dimensional input/outputs. Finally, the authors use their proposed methodology to study how type I and type III interferon Signaling is recognised by cells despite activating the same Signaling effectors.

To my knowledge the methodology that the paper presents is novel and will be of great interest to the wider community of computational biologists. However, I feel that there are a few points that might cause confusion and need to be clarified further.

2.1

1. I believe it would be useful if the authors briefly discussed the following points in greater details:

2.1. i ____

i) **the difficulties involved in calculating FIM**, especially the ones persisting. (In page 4, the authors state: "To the best of our knowledge, this method has not been used to Analyse biochemical signaling, most likely due to technical difficulties in calculating the FIM, which was, to a considerable degree, alleviated by methods only recently developed").

We have added a dedicated section to the supplement to discuss difficulties in calculating FIM, in details. Therein, we also describe how FIM can be calculated in various scenarios. We briefly comment on this in the main paper and refer to the supplement.

In brief, there exist three groups of methods that can serve as an alternative to those used in the paper.

2.1. ii

ii) **how the algorithm scales with number of inputs/outputs** (looking at fig4, I believe the authors already have this information). This discussion will make their claims about the advantages of the method stronger.

As in the previous point, we added as new section to the supplement to discuss the scaling of the algorithm and its potential improvements. In summary, our method is mainly limited by the dimension of the input, as for each combination of input values, the output distribution must be established and the `curse of dimensionality` problem appears. On the other hand, the dimension of output is constrained only by the size model and does not have much influence on the computation cost. Therefore, in practice it would be possible to estimate channel capacity:

- for 1-dimensional input: models with hundreds of species with their dynamical profiles
- for 2-dimensional input: either model of similar size as interferon signaling model (couple dozens of species) with dynamics or model with hundreds of species with fixed time-point measurements
- for 3/4-dimensional input: only simple models with up to 10 species
- for >4-dimensional input: possibly some sparse grid methods (or quasi-Monte Carlo sampling) could be used here to achieve modest computation times and limit the number of required evaluation of the model, but much work is needed to tune those approaches to this specific application.

2.2

2. In page 5, the authors state: "**The capacity $C_A^* > 2$ can be interpreted as the potential of a population of N cells to distinguish both identity and quantity of the two IFNs**";. I am not sure if this is a correct interpretation, the capacity is giving you in broad terms the number of resolvable states, but does not say anything about which states these are. Could the authors please comment on this, as this could change the way they discuss results in Figure 4.

The reviewer's reasoning is correct: the capacity itself does not predict which states are resolvable. However, a system with the capacity lower than two bits cannot correctly discriminate between presence and absence of two stimuli. In other words, **the capacity of two bits is a necessary (not sufficient) condition to discriminate between the presence and absence of the two IFNs.**

We confusingly used the word "potential" meaning that the necessary condition is satisfied. **The paragraph (beginning of page 6 in the current version) have been re-written to clearly state that we meant the sufficient condition.**

2.3

–

3. In figure 6, the authors state: "This reveals interesting, if not surprising, insight. Discriminability at the population level is achieved even with minor differences in kinetic rates, despite high noise levels. Also, **discriminability at the single cell level is not necessary for discriminability at the population level**".

I'm not sure if this is surprising at all. Isn't it what you intuitively expect: the more cells you look the more power to resolve inputs. Also, the authors need to keep in mind that they are not calculating 'discriminability' at the single cell level exactly, but approximating it. There needs to be some (small) 'discriminability' at the single cell level for discriminability at the population level (assuming there are not interaction between cells). There needs to be some (small) 'discriminability' at the single cell level for discriminability at the population level (assuming there are not interaction between cells).

We do not insist it is surprising, especially when thinking how this can be mathematically achieved. Moreover, we agree with the reviewer's line of reasoning: "There needs to be some (small) 'discriminability' at the single cell level for discriminability at the population level". **However, we think it is not straightforward** either, especially when thinking about the biological aspect of this phenomenon.

The model reveals the possibility that two ligands, IFNs in our case, could have different physiological effects (be effectively recognised) with very minor effects on single cells. One would naturally think that if the effect on a single cell is minor the effect on a population is also minor.

Moreover, if confirmed experimentally, our finding could have further implications about which we do not feel competent to write, maybe except one. Discriminability at the population level can be achieved with very minor kinetic differences. Therefore, if tissues or organism can transfer information effectively without discriminability at the single cell level, then the selective pressure to lose cross-reactivity between different components of a signaling pathway is low. **This would contribute to the explanation of highly cross-wired architecture of signaling pathways, especially in multicellular organisms.**

We have substantially rewritten interpretation of the results of Fig. 4 along the lines suggested by the reviewer.

2.4

–

4. **How should one interpret the fact that the asymptotic capacity could go negative?** Since the authors are using it to approximate information capacity (a positive quantity) of single cells, doesn't it make more sense to set it to zero wherever it goes negative?

This issue has been separately discussed in front of this file in the **Main issue section**.

Reviewer #3 (Remarks to the Author):

The authors present an approach to quantify the information transmitted in signaling pathways. To this end, they make use of a series of well-known concepts such as Shannon's information theory and channel capacity, the Blahut-Arimoto algorithm and the Fisher information matrix. They extend this concept by applying it not just to one cell but to a larger number N of cells. They show that the joint channel capacity for N cells can be expressed by the asymptotic information capacity (using the Fisher information matrix) plus a value proportional to $\log_2(N)$.

They apply this concept to two simple or simplified models, a generic receptor activation model and a model of the JAK-STAT pathway. In these models they vary some of the parameter values, but also the type of input distribution. They simulate the models and obtain values for channel capacities.

The manuscript presents an interesting contribution to the ongoing discussion about stochasticity in signaling systems and how this influences their capacity to precisely interpret input signals in terms of gene regulation as output.

From my point of view, a number of issues remain.

3.1

The authors introduce the joint channel capacity for N cells and draw a number of conclusions w.r.t. the response of a cell population instead of single cells. Specifically, they conclude that the population can correctly decode information even in cases where single cells cannot due to high noise or inappropriate receptor kinetics. While the mathematical formalism is straight,

It is unclear to me how the population should be able to make correct decisions based on low capacity in single cells if those don't communicate. Specifically, there is no evidence that this mechanism is realized in cell populations.

Although such mechanisms have not been explicitly described, based on available data on IFNs signaling, we believe, a solid conjecture can be made. At the end of the section *Population level discrimination is possible even at high noise and with minor kinetic differences* we describe how population of cells should be able to make informed decisions based on low capacity of individual cells. Below we elaborate in more detail.

Example 1

For illustration, consider the expression of IFNs induced genes as a downstream output. Both IFNs induce expression of hundreds of genes, including several chemokines from CXCL and CCL family (see references [R3-1, R3-2, R3-3 below]). Specifically, it has been shown that temporal profiles of CXCL10 expression differ in response to both considered IFNs [R3-1]. One of the main function of these chemokines is to attract different types of leukocytes to a site of infections. Therefore, concentration and timing of these chemoattractants can be seen as a decision of cellular population, regarding which and how many leukocytes are needed at a given time.

If the IFN sensing capacity of an individual cell is low the chemokine expression can be only vaguely controlled by IFNs levels. The tissue level of a chemokine is, however, an average expression of neighboring cells, and the resulting concentration is a compromise of a population. Therefore, the high joint capacity of N cells may lead to a finely tuned expression of the chemokine due to averring of the secretion in the extracellular space.

The above constitutes one of several possible example that illustrate how the population should be able to make correct decisions based on low capacity in single cells. The example is in line with current understanding of IFN signaling, specifically with references R3-1, R3-2, R3-3.

The above example does not require cell-to-cell communication, *per se*, *i.e.* although the excretion of the chemokine is a form of information exchange but it does not feed back to IFN signalling

Example 2

As the reviewer points out, it is much easier to imagine how the population could make correct decisions based on low capacity in single cells if those communicated. The information would need to be shared and a compromise made. This would not only allowed a population to make an informed

decision but individual cells could make informed decisions too. It is well known that IFNs induce their own expression (see references [R3-4, R3-5, R3-6]). Therefore one can also imagine that the following scenario is plausible.

An initial stimulus leads to subsequent rounds of cell-to-cell communication through paracrine signaling. Low individual capacity implies that initially the stimulus is recognized with low precision. In subsequent rounds of paracrine communication, information is exchanged between cells, and as a result, the initial stimulus may lead to finely tuned responses of individual cells at later times.

The above mechanisms appear to be plausible. In light of our model predictions, experimental validation of these mechanisms would be highly valuable for reconciling highly stochastic realm of signaling at the single cells with finely-tuned tissue and organismal responses.

References

[R3-1] Bolen, Christopher R., Siyuan Ding, Michael D. Robek, and Steven H. Kleinstein. "Dynamic expression profiling of type I and type III interferon-stimulated hepatocytes reveals a stable hierarchy of gene expression." *Hepatology* 59, no. 4 (2014): 1262-1272.

<https://www.ncbi.nlm.nih.gov/pubmed/23929627>

Specifically, Figure 4 and 5.

[R3-2] Garcin, Geneviève, Yann Bordat, Paul Chuchana, Danièle Monneron, Helen KW Law, Jacob Piehler, and Gilles Uzé. "Differential activity of type I interferon subtypes for dendritic cell differentiation." *PLoS one* 8, no. 3 (2013): e58465.

<http://journals.plos.org/plosone/article?id=10.1371/journal.pone.0058465>

[R3-3] Bauer, Jason W., Emily C. Baechler, Michelle Petri, Franak M. Batliwalla, Dianna Crawford, Ward A. Ortmann, Karl J. Espe et al. "Elevated serum levels of interferon-regulated chemokines are biomarkers for active human systemic lupus erythematosus." *PLoS medicine* 3, no. 12 (2006): e491.

<http://journals.plos.org/plosmedicine/article?id=10.1371/journal.pmed.0030491>

[R3-4] Ivashkiv, Lionel B., and Laura T. Donlin. "Regulation of type I interferon responses." *Nature reviews Immunology* 14, no. 1 (2014): 36.

<https://www.ncbi.nlm.nih.gov/pubmed/24362405>

[R3-5] Hertzog, Paul J., and Bryan RG Williams. "Fine tuning type I interferon responses." *Cytokine & growth factor reviews* 24, no. 3 (2013): 217-225.

<https://www.ncbi.nlm.nih.gov/pubmed/23711406>

Specifically, Figure 1 therein.

[R3-6] Gough, Daniel J., Nicole L. Messina, Linda Hii, Jodee A. Gould, Kanaga Sabapathy, Ashley PS Robertson, Joseph A. Trapani et al. "Functional crosstalk between type I and II interferon through the regulated expression of STAT1." *PLoS biology* 8, no. 4 (2010): e1000361.

<https://www.ncbi.nlm.nih.gov/pubmed/20436908>

Specifically, Figure 8 therein.

3.2

For their simple receptor activation model, they use the small noise approximation. Given that this system is small, I wonder why they don't compare to full stochastic simulation, e.g. with Gillespie algorithm as had been done earlier by other authors. That would allow proper assessment of the system also for larger noise as is assumed to be the case in signaling systems.

We believe we have done our comparison precisely as the reviewer suggests and we believe that any confusion results from lack of clarity and brevity of our description in the main paper.

We have compared the asymptotic capacity C^*_A and the small noise capacity C^*_{SN} with the exact capacity C^*_1 . The latter was calculated using the Blahut-Arimoto algorithm, most accurate available method, for the complete model. The probabilities for the complete model were analytically calculated so we were even more exact than Gillespie algorithm. What is shown in Figure 1 is the deviation in % of the C^*_A and C^*_{SN} from C^*_1 . The considered noise levels were very high. Precisely, the standard deviation of the false ligand concentration were varied up to 50 at receptor's $K_d=1$. Therefore, the false ligand could easily saturate the receptor. High noise is reflected in the capacity close to 0 (Figure S1).

We have reformulated the description in the main paper to provide a clearer explanation that hopefully unambiguously describes how the method was tested.

3.3

—
Again, for this receptor activation model, I don't see why it should be justified to consider that the **cognate signal should be without noise**. It should come with comparable noise levels as the non-cognate signal. What is presented appears as an unjustified simplification of the model.

Similarly to the previous point, we believe, we have performed precisely as the reviewer suggests. The cognate/true ligand must vary similarly as the non-cognate ligand for justified calculation of capacity. This is in fact enforced by the definition of capacity that considered the model input, in our case the true ligand concentration to vary over possibly large range of values. **The concentration of the cognate ligand varied in our model from 3 to 4 orders of magnitude according to optimal input distributions visualized in Fig. S3.**

3.4

—
Most surprising, the authors present simulations with **C^*_A reaching values smaller than zero**. That even comes without any explanation. However, in my humble opinion, this is complete nonsense. Information is by definition non-negative. Hence C^*_A must also be non-negative. If this cannot be ensured it is not useful as a measure for information capacity.

This issue have been separately discussed in front of this file in the **Main issue section**.

Reviewers' comments:

Reviewer #2 (Remarks to the Author):

The authors have responded to all questions/comments thoroughly and have revised their manuscript appropriately. I personally believe that their work makes a novel and interesting contribution to the field of computational and systems biology.

PS I thought that the phrase "All models are wrong, some models are useful" was due to G. Box and not Rao, perhaps I'm wrong.

Reviewer #3 (Remarks to the Author):

The optimization of information transmission is a problem of great interest in the biology of transcriptional regulation. The authors propose an approximation to estimate the optimal input distribution for a given channel which is based on the estimation of the Fisher information matrix (FIM), and on an analytical limit which holds under quite general conditions.

They first test this procedure on a receptor-ligand model, comparing it with the full numerical solution and with the small-noise approximation (SNA).

With their rebuttal, the authors present a revised version of their manuscript. Here, they tried to explain first of all the major problem with the first version, i.e. the occurrence of negative values of information.

It seems that their FIM based method is more detailed than the SNA mainly because they do not restrict themselves to the Gaussian approximation for the input-output conditional distributions, therefore the FIM method is expected to outperform the SNA especially for multimodal input-output conditional distributions. Nevertheless we notice that in real data analysis the Gaussian assumption is made in order not to overfit data, this being often the case in the estimation of input-output conditional distributions from transcriptional data. Their method can still be used for optimizing channel capacity in computational models, the only time limitation being the accuracy in the estimation of the FIM.

Once the Jeffrey prior is estimated, they use it in the supplementary materials also for its corresponding channel capacity (without asymptotic limit) $C^*_{\{JP\}}$, and they state that it is computationally heavier than considering just C^*_A , this being because one has to estimate the conditional probabilities $P(Y|X)$. Is not $P(Y|X)$ already considered for the estimation of the FIM? Is the integral in equation 12 (suppl materials) a long computation when the FIM has already been estimated? These points should be discussed, otherwise it looks more convenient to just use $C^*_{\{JP\}}$.

In figure S2, C^*_A is sometimes more accurate than $C^*_{\{JP\}}$. This is unintuitive and should be discussed as well.

In the interferons model, the channel capacity is estimated only with C^*_A and not with the small-noise approximation. The comparison with the SNA should be done also there, since the aim of the paper is to propose C^*_A as a useful approximation to be compared with the widely used SNA.

Further points:

The authors claim that their manuscript is the first to deal with information processing with multiple inputs/outputs. That's not true, compare for example Metha et al., 2009 (MSB) or Waltermann et al., 2011 (BBA).

To explain the decision making on the population level, the author use the illustrating example of IFN signaling and relate it to potential "averring of the secretion in the extracellular space". That would be an additional input not included in the previous considerations and not fit into the presented concept.

We thank reviewer #3 for the further comments that allowed us to clarify certain ambiguous statements of the manuscript and supplement. Below we respond and describe how the manuscript and the supplement were changed.

3.1 _____

Once the Jeffrey prior is estimated, they use it in the supplementary materials also for its corresponding channel capacity (without asymptotic limit) $C^*_{\{JP\}}$, and they state that it is computationally heavier than considering just C^*_A , this being because one has to estimate the conditional probabilities $P(Y|X)$. Is not $P(Y|X)$ already considered for the estimation of the FIM? Is the integral in equation 12 (suppl materials) a long computation when the FIM has already been estimated?

These points should be discussed, otherwise it looks more convenient to just use $C^*_{\{JP\}}$.

The point, which approximation is more convenient, is indeed very interesting. We have debated this question in detail prior to and during the writing the manuscript. Please note that $C^*_{\{JP\}}$ was presented in the supplement already in the initial submission. Below we explain in more detail why we have chosen to focus on C^*_A rather than $C^*_{\{JP\}}$.

- **Computational efficiency.**

It is true that $P(Y|X)$ is needed to compute FIM. However, an additional integration of the $\log()$ term of the MI formula with respect to Y for each value of the input, X , can be problematic, and costly, especially, if X and Y are multidimensional.

Moreover, for many distributions, including exponential family distributions, FIM can be expressed by an explicit formula or in terms of derivatives of moments of $P(Y|X)$, as highlighted in Section 6 of the Supplement. Given that C^*_A depends solely on FIM integration over highly dimension space of the output Y can be avoided. In case of $C^*_{\{JP\}}$ calculation of FIM and integration over highly dimension space of the output, Y is necessary.

Given the above computational advantages of C^*_A can be significant.

- **Conceptual simplicity**

In our view, C^*_A elegantly links Shannon and Fisher information. It explains how signaling fidelity, measured as standard deviations of signal estimates, relates to the overall sensing capacity. It is, therefore, in our view, intuitive to apply.

- Clear interpretation

$C^*_{\{A\}}$ has a clear and rigorous interpretation of an asymptotic information capacity. In case of $C^*_{\{JP\}}$ the input distribution is asymptotic, whereas the sensing system, i.e., Individual cell, is not. Therefore, to present a focused storyline, without distracting details, we have chosen $C^*_{\{A\}}$ as a focal point of our paper.

We have augmented the explanation of subsection 1.5 of the SI with the above clarifications.

3.2 _____

In figure S2, C^*_A is sometimes more accurate than $C^*_{\{JP\}}$. This is unintuitive and should be discussed as well.

The fact that C^*_A is sometimes more accurate than $C^*_{\{JP\}}$ may seem indeed non-intuitive. However, it can be explained by the following argument.

In the derivation of $C^*_{\{JP\}}$ only one approximation is used, i.e. of the optimal input distribution (Eq. 12, SI). On the other hand, In the derivation of $C^*_{\{A\}}$ two approximations are used, i.e. of the information capacity formula (Eq. 10, SI) and of the optimal input distributions (Eq. 11, SI).

Approximations can either over- or underestimate a true value. If two approximations are used concurrently to approximate a true value, each approximation may happen to have an opposite direction. If this is the case, **the inaccuracies of two approximation will partly cancel out.** Therefore, the seemingly less accurate approximation, C^*_A , can have in certain scenarios higher accuracy than $C^*_{\{JP\}}$.

We have included an explanation of why $C^*_{\{A\}}$ may in certain scenarios be a more accurate approximation of C^* than $C^*_{\{JP\}}$ in subsection 2.2 of the supplement.

3.3 _____

In the interferons model, the channel capacity is estimated only with C^*_A and not with the small-noise approximation. The comparison with the **SNA should be done also** there, since the aim of the paper is to propose C^*_A as a useful approximation to be compared with the widely used SNA.

To the best of our knowledge and further checks, the available extensions of the original version of the SNA [1], specifically these of [2-4], do not account for scenarios with multiple inputs and outputs. Moreover, generalizations that account for several outputs are not straightforward.

To perform the comparison, we would need to develop a generalized version of the SNA. Moreover, the exact value of the capacity cannot be computed for the interferon model due to lack of suitable methods. Therefore, a benefit of the comparison of C^*_A with capacity calculated with some generalized version of the SNA does not appear, in our view, to contribute significant value to our manuscript.

The lack of methods that can deal efficiently with multiple input/output models was precisely the reason why we compared all methods using a simple test model and, further, considered a model for which using existing methods is problematic, if not prohibitive.

[1] Tkačik, Gašper, Curtis G. Callan, and William Bialek. "Information flow and optimization in transcriptional regulation." *Proceedings of the National Academy of Sciences*(2008).

[2] Dubuis, Julien O., Gašper Tkačik, Eric F. Wieschaus, Thomas Gregor, and William Bialek. "Positional information, in bits." *Proceedings of the National Academy of Sciences* (2013).

[3] Tkačik, Gašper, Julien O. Dubuis, Mariela D. Petkova, and Thomas Gregor. "Positional information, positional error, and read-out precision in morphogenesis: a mathematical framework." *Genetics* (2014).

[4] Crisanti, Andrea, Andrea De Martino, and Jonathan Fiorentino. "Statistics of optimal information flow in ensembles of regulatory motifs." *Physical Review E* (2018).

3.4 _____

Further points:

The authors claim that their manuscript is the first to deal with information processing with multiple inputs/outputs. That's not true, compare for example Metha et al., 2009 (MSB) or Waltermann et al., 2011 (BBA).

We have not claimed that we are the first to deal with deal with information processing with multiple inputs/outputs. We claim that our method is first that can efficiently calculate information capacity for systems with multiple inputs and outputs.

We have scrutinized or removed phrasing, all over the manuscript, that could be ambiguous and suggest that we are first to deal with systems with multiple inputs/outputs, which is, of course, not the case. Also, we have included that above valuable references that highlight the need for methods to deal with multiple inputs and output systems.

3.5 _____

To explain the decision making on the population level, the author use the illustrating example of IFN signaling and relate it to potential “averring of the secretion in the extracellular space”. That would be an additional input not included in the previous considerations and not fit into the presented concept.

The chemokines of CXCL and CCL family are the downstream targets of the STAT1/STAT1 and STAT1/STAT2 dimers. Its expression in the single cell is considered to be controlled by the nuclear level of the above homo- and heterodimers. Upon expression, these chemokines are secreted into extracellular space. **Therefore, their inter-cellular concentrations are controlled jointly by a large number of cells. In consequence, the chemokine concentration depends on the information encoded in nuclear levels of the STAT1/STAT1 and STAT1/STAT2 dimers in multiple cells. The above does not require any additional input for the interferon signaling system.**

Therefore, in our view, the example fits into the presented concept of collective information capacity of interferon signaling.

We have added further clarification to the Results section to avoid possible confusion.

REVIEWERS' COMMENTS:

Reviewer #3 (Remarks to the Author):

We feel that the authors have carefully dealt with our concerns and improved their manuscript.